# A smart tele-cytology point-of-care platform for oral cancer screening

Sumsum Sunny[1,2,3,4☯], Arun Baby[4☯], Bonney Lee James[2☯], Dev Balaji[4], Aparna N. V.[4], Maitreya H. Rana[4], Praveen Gurpur[5], Arunan Skandarajah[6], Michael D'Ambrosio[6], Ravindra Doddathimmasandra Ramanjinappa[2], Sunil Paramel Mohan[7], Nisheena Raghavan[8], Uma Kandasarma[9], Sangeetha N.[10], Subhasini Raghavan[10], Naveen Hedne[1], Felix Koch[11], Daniel A. Fletcher[6], Sumithra Selvam[12], Manohar Kollegal[5], Praveen Birur N.[1,10], Lance Ladic[13], Amritha Suresh[1,2], Hardik J. Pandya[4]*, Moni Abraham Kuriakose[1,2]*

1 Head and Neck Oncology, Mazumdar Shaw Medical Centre, NH Health city, Bangalore, India, 2 Integrated Head and Neck Oncology Program (DSRG-5), Mazumdar Shaw Medical Foundation, NH Health city, Bangalore, India, 3 Manipal Academy of Higher Education, Manipal, Karnataka, India, 4 Biomedical and Electronic ($10^{-6}$-$10^{-9}$) Engineering Systems Laboratory, Department of Electronic Systems Engineering, Indian Institute of Science, Bangalore, India, 5 Siemens Healthcare Pvt Ltd, Bangalore, India, 6 Department of Bioengineering & Biophysics Program, University of California, Berkeley, California, United States of America, 7 Department of Oral and Maxillofacial pathology, Sree Anjaneya Dental College, Kozhikode, Kerala, India, 8 Department of Pathology, Mazumdar Shaw Medical Centre, NH Health city, Bangalore, India, 9 Department of Oral and Maxillofacial Pathology, KLE Society's Institute of Dental Sciences, Bangalore, India, 10 Department of oral medicine and radiology, KLE Society's Institute of Dental Sciences, Bangalore, India, 11 University of Mainz, 55099, Mainz, Germany, 12 Division of Epidemiology and Biostatistics, St. John's Research Institute, St. John's National Academy of Health Sciences, Bangalore, India, 13 Siemens Healthineers, Malvern, Pennsylvania, United States of America

☯ These authors contributed equally to this work.
* hjpandya@iisc.ac.in (HJP); makuriakose@gmail.com (MAK)

**Data Availability Statement:** All relevant data are within the manuscript and its Supporting Information files.

**Funding:** This work was supported by the Wellcome Trust/DBT India Alliance Fellowship [IA/

## Abstract

Early detection of oral cancer necessitates a minimally invasive, tissue-specific diagnostic tool that facilitates screening/surveillance. Brush biopsy, though minimally invasive, demands skilled cyto-pathologist expertise. In this study, we explored the clinical utility/efficacy of a tele-cytology system in combination with Artificial Neural Network (ANN) based risk-stratification model for early detection of oral potentially malignant (OPML)/malignant lesion. A portable, automated tablet-based tele-cytology platform capable of digitization of cytology slides was evaluated for its efficacy in the detection of OPML/malignant lesions (n = 82) in comparison with conventional cytology and histology. Then, an image pre-processing algorithm was established to segregate cells, ANN was trained with images (n = 11,981) and a risk-stratification model developed. The specificity, sensitivity and accuracy of platform/ stratification model were computed, and agreement was examined using Kappa statistics. The tele-cytology platform, Cellscope, showed an overall accuracy of 84–86% with no difference between tele-cytology and conventional cytology in detection of oral lesions (kappa, 0.67–0.72). However, OPML could be detected with low sensitivity (18%) in accordance with the limitations of conventional cytology. The integration of image processing and development of an ANN-based risk stratification model improved the detection sensitivity of malignant lesions (93%) and high grade OPML (73%), thereby increasing the overall accuracy by 30%. Tele-cytology integrated with the risk stratification model, a novel

RTF/15/1/1017] awarded to Sumsum Sunny (https://www.indiaalliance.org/fellowsprofile/dr-sumsum-sunny–270). Additionally, funders provided support in the form of salaries for authors [DBT Alliance fellowship: SPS; Siemens Healthcare Pvt Ltd and Siemens Healthineers: LL, PG, MK], but did not have any additional role in the study design, data collection and analysis, decision to publish, or preparation of the manuscript. The specific roles of these authors are articulated in the 'author contributions' section.

**Competing interests:** LL, PG, and MK are paid employees of Siemens Healthcare Pvt Ltd and Siemens Healthineers at the time that a major part of this work was carried out. Some of the Siemens employees are also owners of Siemens shares. Dr Fletcher is co-founder of CellScope Inc., a company commercializing a cell-phone based microscope. CellScope Inc had no involvement with the study described in the manuscript. This does not alter our adherence to PLOS ONE policies on sharing data and materials.

strategy established in this study, can be an invaluable Point-of-Care (PoC) tool for early detection/screening in oral cancer. This study hence establishes the applicability of tele-cytology for accurate, remote diagnosis and use of automated ANN-based analysis in improving its efficacy.

## Introduction

Oral cancer accounts for 30% of cancer-related death in low and middle-income countries [1]. Risk stratification of oral potentially malignant lesions (OPML) and early malignant lesions may help to initiate therapeutic intervention and may improve the prognosis. Biopsy and histopathology-based grading of OPML is the current standard of care. However, due to the invasive nature of biopsies and lack of related expertise, this is neither feasible nor readily utilized as a screening tool. These issues are owed to the scarcity of trained specialists such as pathologists or surgeons in low-resource-settings. The studies shows that less than 65% of primary care centres have access to reliable pathology services in low-middle income countries[1–3]. Hence, a tele-cytology platform that provides reliable, remote connectivity to frontline health workers (FHW) and specialists may improve early detection of oral cancer.

Oral cytology, is considered an effective tool for the large-scale screening of high-risk populations [4, 5]. Telemedicine platforms have been used in cytology with proven benefits in remote diagnosis of cervical [6], lung [7, 8], breast [9, 10], and thyroid malignancies [11, 12]. Additionally, automated analysis of tele-cytology images using machine learning is an aspect that will impart a Point-of-Care (PoC) applicability to the system and is increasingly required wherein additional skilled manpower is not available. The applications of ANN have been previously explored for the classification of oral diseases [13] and cytopathology diagnosis of cervical [14–16], breast[17, 18], and blood malignancies [19]. Combining tele-cytology along with ANN will be a step towards translating the platform into a point of care application in oral cancer.

The platform used in this study (Fig 1) was an iPad tablet-based version of the "CellScope" mobile microscope [20], capable of automated focusing of cells, scanning of cytology slides, and uploading the captured images to a specialized web-based server. Our initial study showed the feasibility of tele-cytology in connecting FHW with pathologists [20]. In this study, we explored the clinical utility/efficacy of this portable, automated system in combination with Convolutional Neural Network (CNN) for classification of atypical cells [21] and subsequent training of the ANN, Inception V3 architecture [22]. We hypothesize that integration of ANN with the tele-cytology platform may improve risk stratification of OPML. This pilot study validates the risk stratification model prior to implementation in a low resource setting.

## Materials and methods

### Study population and data acquisition

The study was carried out among subjects attending the out-patient clinics of Department of Oral Medicine and Radiology, KLES Institute of Dental Sciences, Bangalore and Head and Neck Oncology Department, Mazumdar Shaw Medical Center, Bangalore, India for a period of 24 months from October 2014 to September 2016. The Narayana Health Medical Ethics Committee has approved the study (NHH/MEC-CL-2014/222) and the subjects clinically diagnosed with OPML and malignant lesions were recruited, while those who did not consent

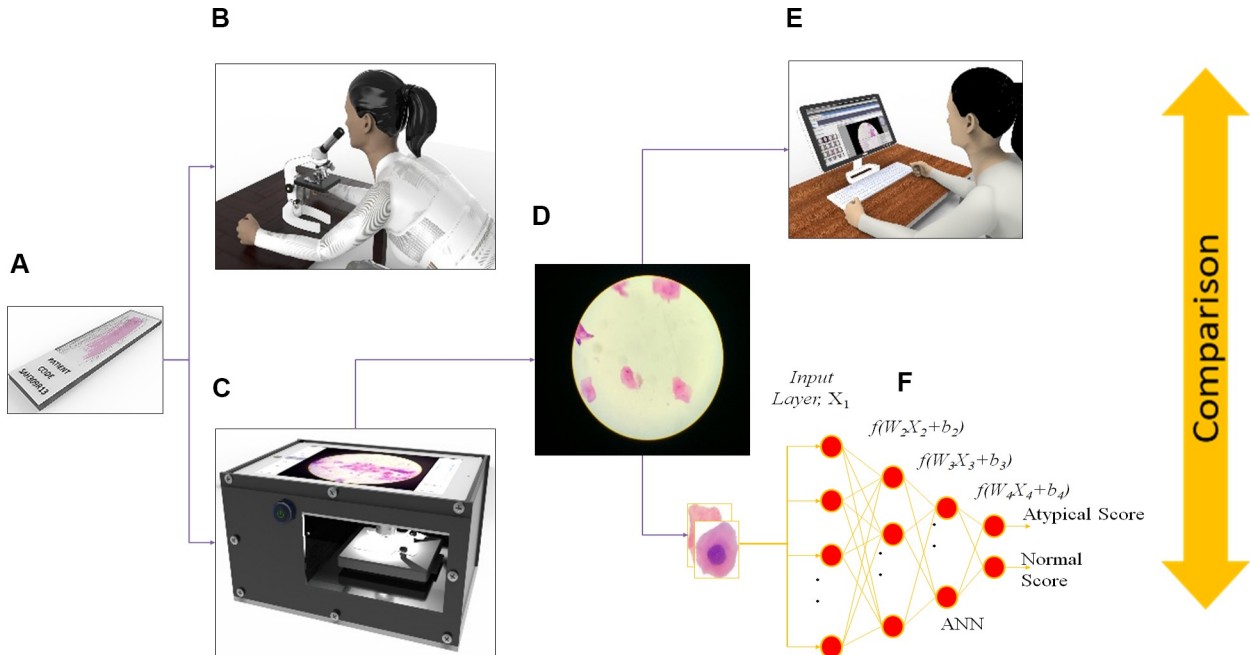

**Fig 1. Study design.** Microscopic slides were prepared (a) using liquid based cytology and slides were reviewed by (b) conventional direct microscopy. Images were captured using CellScope (c) and sent to remote server (d). Tele-cytology diagnosis (f) were performed by pathologist. Image pre-processing algorithm were developed, and ANN based cytology diagnostic platform were developed (g) and validated. Conventional cytology diagnosis, tele-cytology diagnosis and ANN based diagnosis were compared with histopathology.

and/or had undergone biopsy previously were excluded from the study. Sample collection and slide preparation was carried out as reported previously [20].

High resolution cytology images (2592 x 1936) of 24-bit depth with optical resolution of 1.9 pixel/micrometre (optical resolution 200X) were captured using the CellScope as described previously [20] (S1A and S1B Fig). This platform captured the fields (100–125) in a raster scan pattern, the images were connected to the clinical/demographic data of the patients entered into the iPad application. (S2A Fig). For remote diagnosis, the pathologists utilized a custom, web portal interface for blind review of images [20], which enabled them to zoom into the image and visualize the morphological features without loss of resolution (S2B–S2E Fig). The overall quality of the images of each subject (n = 32, 3200 images) were assessed by recording the following parameters: i) overall image quality (good, adequate and poor), ii) diagnostic capability (diagnostic and non-diagnostic) and iii) time taken for diagnosis (<5 minutes, 5–10 minutes and >10 minutes).

## Pathological review pipeline

Two board-certified pathologists conducted a blinded, review of the slides using tele-cytology and direct-microscopy and documented the following features: multi-nucleation, mitotic figures, prominent nucleoli, altered nuclear-cytoplasmic ratio, hyper-chromatic nucleus, irregular nuclear membrane and any additional features as per their discretion. These features, adapted from the oral/pharyngeal cytological scoring system [23], were confirmed after a consensus from both the pathologists.

**Tele-cytology diagnosis (remote diagnosis).** The two pathologists logged onto the secure server, demarcated specific cells in the uploaded images as region of interest (ROI) and

documented its abnormal features. The subjects were annotated as '*most likely positive*' or '*most likely negative*' or '*unable to interpret*'.

**Cytology diagnosis (direct microscopy).** Cytology diagnosis by direct microscopy was diagnosed by indicating the presence (1)/absence (0) of atypia on the basis of the cytological characters mentioned above. The cytology score was calculated for each case by the sum of the individual scores and compared with the histological diagnosis.

**Histopathology diagnosis.** The specimens were evaluated by routine histopathology and diagnosis was reported according to WHO criteria [24, 25]. Hyperkeratosis with epithelial hyperplasia and mild dysplasia were considered as low-grade dysplasia (LGD), while moderate dysplasia, severe dysplasia and *carcinoma-in-situ* were categorized as high-grade dysplasia (HGD) based on the binary system of classification [26].

### Automated diagnosis and ANN workflow

A subset of subjects (n = 60) was selected for development and validation of a risk stratification model. Tele-cytology images were segmented to detect and segregate cells and labelled as atypical or normal by a pathologist (S3 Fig). These images were used for training an existing ANN (Inception V3) using transfer learning. All programming related to the ANN was implemented in Python (Google's TensorFlow library; https://github.com/tensorflow/tensorflow). The learning was done for 4000 epochs with a learning rate of 0.01, wherein 90% of the dataset were used for training and 10% for validation. After training, each of the patients' extracted cells was fed to the ANN to generate a score between 0 and 1. (0: most likely normal, 1: most likely atypical). The results of individual cells were aggregated for each patient; cells with scores above 0.5 (out of 1) were taken as atypia. The percentage of atypical cells, mean score of all cells and the mean score of atypical cells of individual patients were calculated. A classification learner model was developed using these values and validated.

### Statistical analysis

The minimum sample size was calculated for screening study[27]. The prevalence of expected neoplastic and high grade dysplastic lesions were approximately 80% [20]. We expected 15% improvement in sensitivity [20, 28], considering the alpha value of <0.05, power of 80% and drop out (50%) the minimum sample size required for the study was 98.Therefore 100 subjects were recruited for the study including OPML and Malignant lesions.

Descriptive statistics were used to summarize details of patient demography, clinical features, and pathological diagnosis Normal distribution of continuous variables were tested using Kolmorgorov-smirnov test [29]. All statistical comparisons between multiple groups were assessed by one-way analysis of variance (ANOVA; Kruskal-Wallis test). Pearson correlation coefficient was used to find the correlation between the variables. The specificity, sensitivity and accuracy of tele-cytology, direct cytology and risk stratification model were computed. McNemar test was used to compare the proportions. The agreement between diagnoses was examined using Kappa statistics [30, 31]. ROC curve analysis was performed to find the cut-off score. P value less than 0.05 was considered statistically significant in all analyses. All statistical analyses were done using SPSS version 23 and MedCalc v14.8.1.

## Results

### Clinical and pathological details of the patients

Subjects recruited for the study were largely from southern states of India, including Karnataka, Tamil Nadu, and Kerala. A total of 100 patients were recruited for the study after written

informed consent. The protocol, merits and confidentiality of the study was explained to all subjects in their native language. The consent form was provided to the patient in their respective language and was signed by them, their physician and the study investigator. Among 100 patients, biopsy could not be performed for 16 subjects, while two were rejected due to low cell number. Eighty-two patients were hence selected for further analysis; majority of the subjects were males (males: 78%; females: 22%) with a mean age of 45.4 years. Eighty-four percent (n = 69) subjects reported at least one of the high-risk habits: chewing, smoking or alcohol consumption. Demographic details were given in S1 Table. Patient consort chart indicating pathological diagnosis is provided (S4 Fig).

## Tele-cytology shows good agreement with direct microscopy

The quality assessment of the tele-cytology images (n = 2880) in the patient cohort (n = 32) indicated that 96% (n = 2765) were of good quality. Assessment of time taken for diagnosis showed that the majority of the patients could be diagnosed within 10 minutes (61%), whereas conventional cytology-based evaluation recorded an average time of 15 minutes for diagnosis. Additionally, the diagnosis of carcinoma could be arrived at with a lesser number of images (n = 20) as compared to dysplasia (n = 100).

Remote diagnosis showed an overall average sensitivity of 81% (85–76) and average specificity of 90% (84–96) in detecting atypical cells (OSCC/HGD Vs LGD) when compared with direct microscopy (gold standard) (Table 1). Tele-cytology diagnosis revealed high PPV (90%; 86.7–93.3) and NPV (82%; 81.1–82.7). There was no significant difference in tele-cytology and conventional cytology diagnosis of either pathologist (McNemar's test; pathologist I p-value: 1; pathologist II p-value: 0.06) with the Kappa value (0.67–0.72; p<0.05) indicating good agreement [30, 31].

Assessment of the efficacy of diagnosis between OPML and malignant lesions (OSCC) separately indicated a higher accuracy of diagnosis in malignant cases with both methods. Among the malignant lesions diagnosed by conventional cytology (35/38); 80–97% were diagnosed by tele-cytology (pathologist I: 97%; pathologist II: 80%), indicating a nearly perfect concordance between the two techniques. However, in the case of HGD, the diagnostic efficacy was only comparable between the two methods. Pathologist I diagnosed 25% (8/32) of HGD as atypia by conventional cytology, out of which 38% (3/8) were detected using tele-cytology. Notably, 3

**Table 1. Sensitivity, specificity and accuracy of Tele-cytology and direct microscopy.**

| Test Vs reference standard | | Pathologist I | | | Pathologist II[a] | | |
|---|---|---|---|---|---|---|---|
| | | Sensitivity | Specificity | Accuracy | Sensitivity | Specificity | Accuracy |
| OSCC / HGD Vs LGD[b] | Tele-cytology Vs direct microscopy | 84.8 | 83.3 | 84.1 | 75.7 | 95.6 | 86.6 |
| | Direct microscopy Vs HP[c] | 61.4 | 75 | 63.4 | 67.9 | 96.6 | 78.1 |
| | Tele-cytology Vs HP | 60 | 75 | 62.2 | 54.7 | 96.6 | 69.5 |
| OSCC Vs LGD | Tele-cytology Vs direct microscopy | 94.7 | 72.7 | 89.8 | 77.8 | 93.3 | 84.9 |
| | Direct microscopy Vs HP | 92.1 | 72.7 | 87.76 | 92.1 | 96.4 | 93.9 |
| | Tele-cytology Vs HP | 94.7 | 72.7 | 89.8 | 76.3 | 96.4 | 84.85 |
| HGD Vs LGD | Tele-cytology Vs direct microscopy | 45.5 | 87.5 | 76.7 | | | |
| | Direct microscopy Vs HP | 25 | 72.7 | 37.2 | | | |
| | Tele-cytology Vs HP | 18.8 | 72.7 | 32.6 | | | |

[a]Pathologist II couldn't detect atypical cells in HGD, LGD using cytology.

[b]OSCC = Oral squamous cell carcinoma, LGD = Low grade dysplasia, HGD = High grade dysplasia.

[c]HP = Histopathology diagnosis

cases of HGD missed by conventional cytology were diagnosed using tele-cytology, indicating a comparable overall detection efficacy (19%; 6/32). Pathologist II did not detect any of the HGD lesions by tele-cytology (Table 1).

## Tele-cytology and direct microscopy correlate with neoplastic histology

In the diagnosis of the malignant cases (n = 38) both pathologist, pathologist I (95%, n = 36) and pathologist II (76%, n = 29), showed good sensitivity (76–95%) and specificity (73–96%) in comparison with histology. Tele-cytology diagnosis of HGD lesions (Pathologist I; 32/43) revealed a sensitivity and specificity of 18.8% and 72.7% respectively. The overall efficacy (in detecting OSCC and HGD lesion) of tele-cytology with histology as the gold standard were 62% and 69.5% respectively (Table 1).

The efficacy of conventional cytology using direct microscopy when compared to histology as the gold standard indicated that, as observed in the case of tele-cytology, the discrepancies were in the diagnosis of HGD. Among the 38 neoplastic cases, 92% (n = 35) of cases were diagnosed by both pathologists by direct microscopy. However, pathologist I and II detected atypia in only 25% (n = 8/32) and 7% (n = 1/15) of HGD lesions respectively by direct microscopy, when compared to their individual histological assessment.

To assess the efficacy of the cytology features adapted from the oral and pharyngeal cytological scoring system [23], each of them were individually assigned a score and total manual cytology score calculated for each subject. The most important cytological features sufficient for diagnosis were irregular nuclear membrane, abnormal cell shape and increased nuclear-cytoplasmic ratio (S2C–S2E Fig). Comparison of the cytology score with the histology diagnosis indicated that although these features could significantly delineate malignant patients from those with dysplasia ($p < 0.0001$), they could not demarcate between HGD and LGD ($p = 0.32$).

## Image processing algorithm to obtain individual cells

Each of the tele-cytology images were passed into an image processing algorithm (Fig 2A and 2B, S3 Fig) (ImageJ) wherein the cells were segmented, Field of View (FoV) boundaries detected and the cells masses (S5 Fig) detected based on the highest contrast in the green channel (RGB colour space). In case of presence of clumped cells, a watershed algorithm was used to approximately disconnect them [32].

These images were further analysed with a particle analysis algorithm (ImageJ) and individual cells' region of interests (ROIs) obtained. Each ROI was then cropped and verified against a set of criteria (S5 Fig and S6 Fig) [33]. Finally, these thresholded images were passed through the particle analysis tool to check the presence of a nucleus. The images of cells, which passed these quality checks, were then fed to the neural network for classification (S3 Fig). A total of 11,981 cell images from 60 patients were segmented, and an average of 200 cells per patient were obtained. Each tele-cytology image took less than a second to be segmented into individual cells (Fig 2B).

## ANN scoring and risk stratification model correlate with tissue-specific diagnosis

The training set included cell images (normal: 252; atypical: 280) labelled by the pathologist as atypical (Fig 3A) or normal (Fig 3B) and taken randomly from six subjects (LGD: 3; OSCC: 3). These images were removed from the validation dataset. The images were augmented to generate a larger training set (n = 12,768 augmented images). The ANN (S7 Fig) was trained for 4000 epochs (Fig 2C) with final validation accuracy of 95%. Each cell takes approximately one second to get classified and on average 3 minutes for categorizing all cells of a patient. Cells

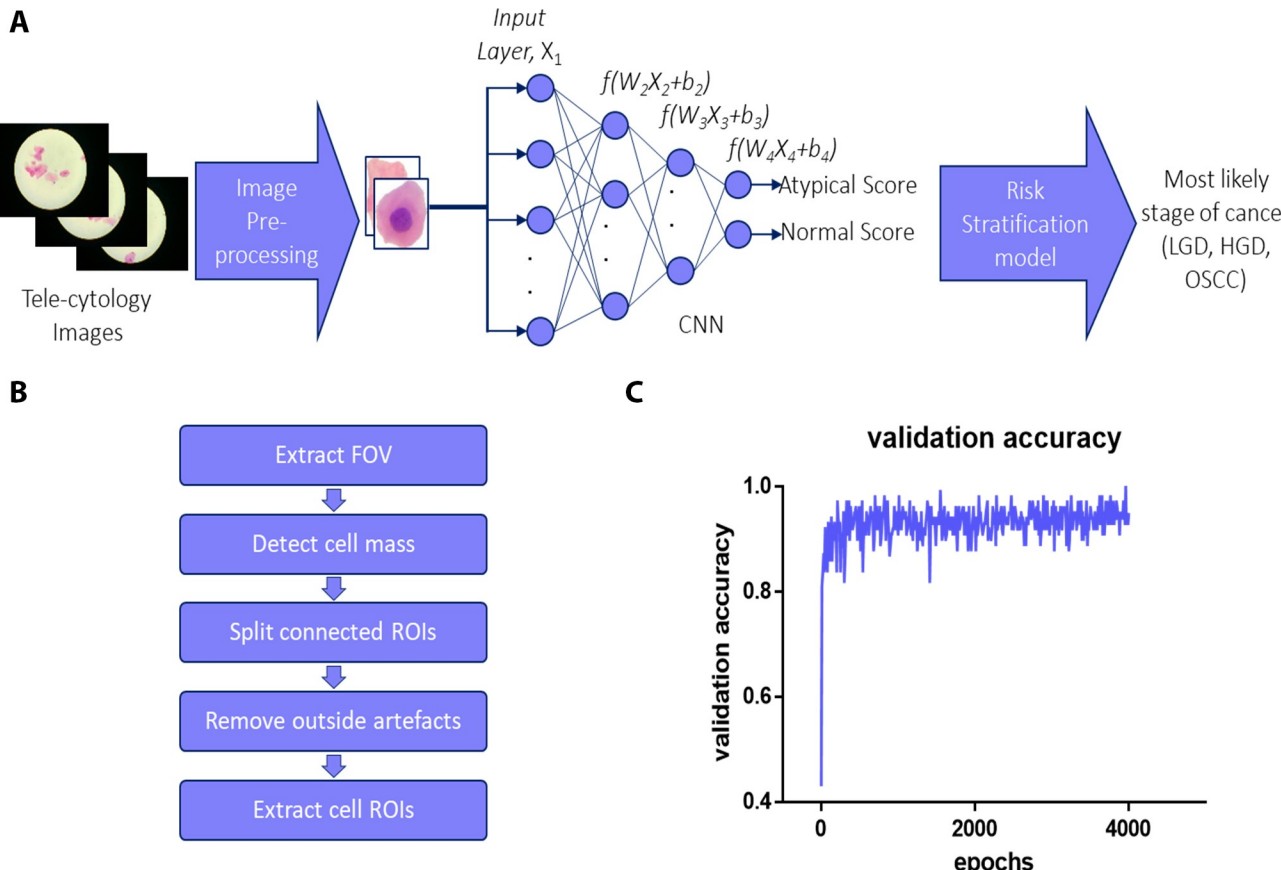

**Fig 2. Workflow of image processing and ANN.** Complete workflow (a) diagram of the automated diagnosis system; The cells are extracted from the tele-cytology images and are fed into a neural network and the values from all the cells in a patient are aggregated and used for developing risk stratification model. Image pre-processing algorithm (b) consisting of Field of View (FOV) extraction from the tele-cytology images, detection of contrasting cellular mass from the background, detachment of connected Region of Interests (ROIs), removing the artefacts outside the ROIs, and extracting cell ROIs. The graph represents validation accuracy during training (epochs = 4,000) of the ANN (c).

that gave scores above 0.5 were assumed to be atypical cells. An increasing abnormality of the cell was correlated with increase in atypical scores (Fig 3C–3G). The percentage of atypical cells, mean score of all cells and the mean score of atypical cells were calculated for each patient by below formula.

$$\text{Percentage of atypical cells} = \frac{\text{Number of cells having score above 0.5}}{\text{Total number of cells}} \times 100 \qquad (1)$$

$$\text{Mean score of atypical cells} = \frac{\text{Sum of scores of atypical cells}}{\text{Number of atypical cells}} \qquad (2)$$

$$\text{Mean score of all cells} = \frac{\text{Sum of score of all cells}}{\text{Total number of cells}} \qquad (3)$$

The average percentage of atypical cells in benign (BNG)/LGD, HGD and OSCC were 14%, 16% and 39%, respectively. Manual cytology score calculated by adding cytological features (Fig 4A), the mean score of all cells (Fig 4B) and percentage of atypical cells (Fig 4C) showed a

| | Atypical score | Histology diagnosis | Sample images | | | | |
|---|---|---|---|---|---|---|---|
| A | Trained | Atypical cell (OSCC) | | | | | |
| B | Trained | Normal cell (Lichen planus) | | | | | |
| C | <0.03 | Benign | | | | | |
| D | 0.3-0.49 | LGD | | | | | |
| E | 0.5-0.69 | HGD | | | | | |
| F | 0.7-.89 | OSCC | | | | | |
| G | >0.9 | OSCC | | | | | |

**Fig 3. A batch of trained and validated cell images.** Images of atypical cells(a) and normal cells (b) used for training the ANN. Cells classified by ANN: cells having atypical score less than 0.3 (c) from benign subjects, cells with atypical score between 0.3 to 0.5 from LGD patients(d), cells with atypical score between 0.5 to 0.7 (e) from HGD patients, cells with atypical score between 0.7 to 0.9 (f) from OSCC patients and cells with atypical score greater than 0.9 (g) from OSCC patients.

statistically significant difference between dysplasia (HGD or LGD) and OSCC (p<0.005), but no significant difference between LGD and HGD (p>0.05). The mean score of atypical cells (Fig 4D) showed statistically significant difference between LGD and HGD (p<0.05). These three parameters were considered to make the risk stratification model.

The risk stratification model included two tests; initially, a model was developed to delineate OSCC from HGD/LGD/Benign using the classification learner module (MatlabR2018a). The mean score of cells and percentage of atypical cells (which showed strong correlation; r = 0.992) of 50% patients (n = 30) were randomly selected for training classification model (Fig 5A). The model was then validated with another 30 patients using holdout validation. Among the linear models (MatlabR2018a classifier documentation) trained and compared (Support Vector Machine (SVM), Random forest, Logistic regression, Linear Discriminant Analysis and K-Nearest Neighbour) [34] (S2 Table), the SVM model gave the best accuracy, with a sensitivity and specificity of 93% (n = 14/15) and 88% (n = 13/15) respectively (Test 1,

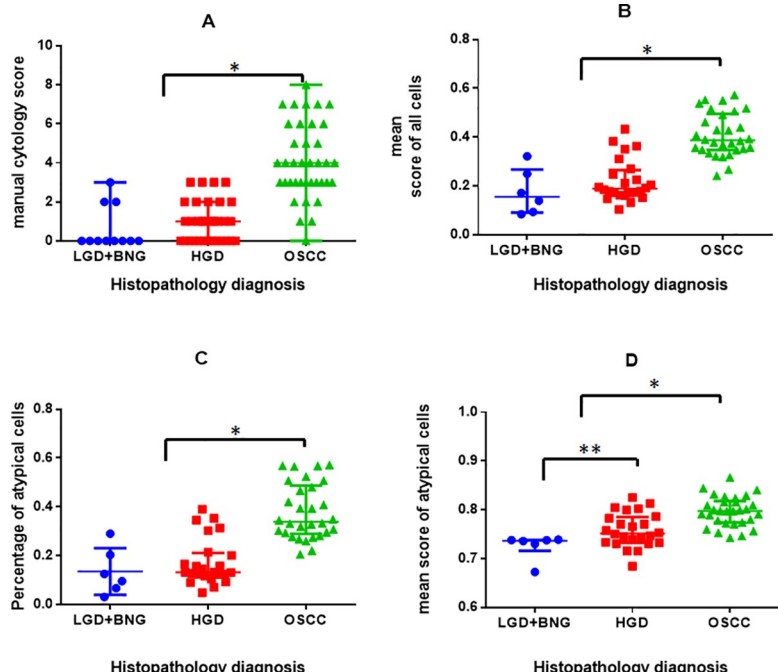

**Fig 4. Distribution of manual and ANN cytology scores.** Box and whisker plot represent (a) cytology score of direct microscopy method (n = 82), OSCC (4.08±1.92) score shows significant difference from (*p<0.005) LGD (0.63±1.12) and HGD lesions (1±1.05). ANN Scoring (n = 60): The mean score of all cells (b) shows statistical significance between dysplasia (HGD, LGD) and OSCC (0.40±0.08, *p<0.005), but does not show significant difference between LGD (0.17 ±0.09) and HGD (0.21±0.08). The percentage of atypical cells (c) OSCC (0.38±0.11) shows significant difference from dysplasia (*p<0.005) but not show significant between HGD (0.17±0.09), and LGD (0.17±0.09). The mean atypical score of atypical cells (cells having score >0.05) (d) in each patient demonstrating statistical significance between dysplasia (HGD, LGD) and OSCC (0.71±0.02, *p<0.005) and also between LGD (0.78±0.03) and HGD (0.76±0.03) (**p<0.05). The mean and standard deviation values are provided in brackets.

Fig 5C). The patients that the model predicted as positive were considered to have a high risk of OSCC, while the patients that predicted as negative were passed through the second test.

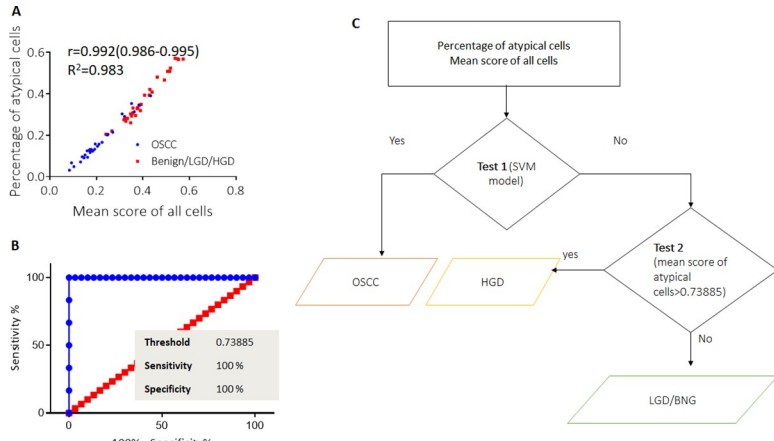

**Fig 5. Risk stratification model.** Scatter plot (a) representing percentage of atypical cell and mean score of all cells (n = 60) showing high correlation (r = 0.992, CI = 0.986–0.995) and these variables used for test 1, in risk stratification model (SVM). The cut-off value of ROC curve analysis (b) in delineating OSCC from LGD were used in risk stratification model as test 2 (c).

**Table 2. Sensitivity and specificity of manual cytology method and risk stratification model.**

| | OSCC Vs HGD/LGD | HGD/ LGD | OSCC/HGD Vs LGD | Accuracy |
|---|---|---|---|---|
| **Cytology Manual Method (n = 30)** | | | | |
| Sensitivity | 87 (13/15) | 25 (3/12) | 59 (16/27) | 60% |
| Specificity | 73 (11/15) | 66 (2/3) | 67 (2/3) | |
| **Risk stratification model (n = 30)** | | | | |
| | Test 1[a] | Test 2[a] | | |
| Sensitivity | 93 (14/15) | 73 (8/11) | 89 (24/27) | 90% |
| Specificity | 88 (13/15) | 100 (3/3) | 100 (3/3) | |

[a]Test 1 and test 2 explained in risk stratification model (Fig 5)

The optimal threshold cut-off value for delineating OSCC from LGD was calculated from the mean score of atypical cells alone (Fig 5B). This cut-off was used to delineate HGD from LGD (Test 2, Fig 5C). The data from the subjects showing positive results were considered most likely to have HGD and OSCC. The negative subjects were considered to have LGD or Benign lesions (Fig 5C). The test gave a sensitivity and specificity of 73% (8/11) and 100% (3/3) respectively for delineating HGD from LGD/Benign. The entire model had an accuracy of 90% in delineating OSCC and HGD from LGD with 89% sensitivity, while the direct microscopy (same cohort, n = 30) had 59% sensitivity with an accuracy of 60% (Table 2).

## Discussion

The estimated delay in diagnosis of oral cancer from the time a patient seeks medical assistance is three months [35]; lack of specialists' expertise at the primary care settings being a major factor. Brush biopsy and cytology, are readily adaptable in primary health care centres [36] and are currently being investigated as a tool for oral cancer risk-stratification [37, 38]. The precision in detection and risk stratification can be further improved using machine learning algorithms, which also detaches the subjectivity of cytology interpretation. In this study, we evaluated the efficacy of a tele-cytology platform for oral cancer screening and developed a risk stratification model using an Artificial Neural Network. The results of the study the clinical efficacy of the platform and an improved accuracy in the diagnosis of OSCC/HGD with integration of ANN indicated that automated image capture/analysis provided the requisite information essential for point-of care remote diagnosis.

We have previously reported the feasibility of remote pathology diagnosis in oral cytology using a mobile and automated tele-cytology device [20], CellScope. This study showed that in addition to the high efficacy in capturing images (96% of images were of diagnostic quality), tele-cytology also showed high sensitivity/specificity in the diagnosis with good agreement (K = 0.68–0.72) with the conventional direct microscopy-based method. Similar ranges of agreement (K = 0.47 to 0.77) have been reported in tele-cytology-based diagnosis of cervical cancer [30, 38]. While the previous tele-cytology studies, necessitated a specialist intervention at the PoC [30, 38, 39], the tele-cytology pipeline detailed in this study enabled remote diagnosis facilitating risk stratification and appropriate triaging of patients by a FHW. The images could be transferred using the mobile cellular network with an adequate resolution of the cellular morphology for accurate interpretation, making it a potential tool in a low-resource setting. This prospective, blinded study using a tele-cytology platform, also showed a high level of concordance with conventional cytology by direct microscopy.

Although cytology could diagnose oral cancer with high efficacy, its ability to detect atypia in HGD lesions was low, which was underscored by the low sensitivity of both tele-cytology

and direct microscopy in patients with HGD. This is in accordance with the poor performance of manual cytology to diagnose HGD reported in multiple studies [40, 41]. The most common cytological features considered for diagnosis in this study, though sufficient for the detection of OSCC (Fig 4A, cytology score >3), were not efficient for detecting HGD. Limitations of the brush biopsy in obtaining cells from the deeper layers due to high keratinization of stratified squamous epithelium [42] might be the primary reason, leading to subjectivity in diagnosis. This platform is hence inefficient in the detection of HGD with current standard cytological features (sensitivity = 25%). These challenges necessitated the use of objective machine learning solutions along with the tele-cytology platform to enable better stratification of patients.

Integration of automated diagnosis in this study included development of new image preprocessing algorithm that segmented individual cells (200-300cells/patient) within each image, removed clumped cells/artefacts. Further improvements in algorithms that can effectively utilise the clumped cells may increase the number of diagnosable images, however advanced segmentation algorithms available are computationally intensive [43–45]. We have not implemented such algorithms in the device keeping in mind that there will be low computational resources available at PoC. The Convolutional Neural Network (CNN) adopted in this study enables direct image input with the filters being trained to extract features automatically [21]. This bypasses the need to have well-defined criteria to detect HGD that incapacitated accurate manual diagnosis. ANN classification is robust in classifying cells that are over-segmented and in this study we used transfer learning, wherein a pre-trained ANN is fine-tuned using the new dataset, a method well-used for training small data set [46]. Inception V3, used in this study, was chosen due to its ability to provide better accuracy while using computational resources effectively [22] and is a network tested in various forms of cancer detection such as in cervix [47], skin [48], breast [49] and lung [50]. Additionally, if a need arises, the automated system allows for the images that are classified as atypical to be sent for remote pathologist review, lowering the network bandwidth required for image transfer and thus improving the throughput and reach of the pathologist.

The risk stratification model developed in this study adopted a sequential mode for patient stratification; it first detected malignancy (sensitivity: 93%) using linear SVMs (with 90% accuracy) and then delineated HGD (sensitivity = 73%). SVMs were shown to have good performance in a previous study for cytopathology-based DNA Index in oral leukoplakia [51]. The differential efficacy in the diagnosis of OSCC and HGD/LGD might be attributed to the presence of a higher percentage of atypical cells with a significantly higher mean score of all the cells imaged (p<0.05). Given this discrepancy in the percentage of atypical cells, in this model, delineation of HGD and LGD was based on the score of atypical cells alone. These criteria could distinguish the two dysplastic groups (p = 0.043). The entire automation involving image segmentation and risk stratification of a patient took around ten minutes making it as fast and appropriate as a point of care, tele-cytology tool.

## Conclusion

The tele-cytology platform evaluated in this study is an effective tool for remote diagnosis since it could successfully retain features of diagnostic value. ANN-based automated diagnosis and risk stratification improved the sensitivity in detection of HGD lesions. This, in turn, increased the overall accuracy of the system by 30% when compared to the manual method. A study with a larger cohort is required to improve the robustness of the system in low resource environments. Nevertheless, this pilot study is a significant effort to improve the accuracy of oral cytology-based risk stratification and for enabling tele-cytology-based point of care diagnosis.

## Supporting information

**S1 Fig. Cellscope slide scanning device.** Front View of the device (**A**) showing the scanning platform and top view showing the iPad mini 2 (**B**) as the user interface.
(TIF)

**S2 Fig. Web interface used in iPad for entering patient information and representative images.** An image, (**A**) captured by Cellscope (200X magnification) with resolution of 2592 x 1936 (72dpi) and cell with irregular nuclear membrane, (**B, blue arrow**). The images were zoomed in to 200% representing cell with irregular nuclear membrane, (**C**), abnormal cell shape, (**D**) and increased nuclear to cytoplasmic ratio, (**E**).
(TIF)

**S3 Fig. Detailed flowchart for image pre-processing.** An Input image, (**A**) was analysed to estimate the Field of View (FOV) and was trimmed to contain only the same, (**B**), once detected, the green channel from RGB image, (**C**) was considered to detect areas of cellular mass owing to its better contrast of cell mass. These images were thresholded, (**D**) to get cellular area. The binarized image is then analysed for extracting high level Region of Interests (ROI), (**E**) that is used to clear the background, (**F**). Images were then water segmented, (**G**) and again analysed for cellular-level ROIs, (**H**). Then each ROI, (**I**) is then extracted as a single image. The red channel, (**J**), is then obtained and thresholded, (**I**) to detect nucleus. The images are then checked with pass criteria and then saved, (**L**) into a folder.
(TIF)

**S4 Fig. Study consort chart.** Distribution of subjects according to clinical and histopathological diagnosis. Eighty subjects were included in the analysis, of which, 43 were OPML and 39 were malignant. For automated diagnosis, 22 subjects were excluded since their images were taken during the developmental stages of the tele-cytology platform and the images were very different from the final set of images. Thus (n = 60) subjects were considered for development and validation of ANN based diagnosis. OPML = Oral Potentially Malignant Lesion, HGD = High Grade Dysplasia, LGD = Low Grade Dysplasia, OSCC = Oral Squamous Cell Carcinoma.
(TIF)

**S5 Fig. Detailed flow chart for FOV estimation.** The algorithm finds out 3 pixels at the edge of the FOV in the input image, (**A**), Assuming the FOV is circular, the equation of circle $(x-p)^2+(x-p)^2 = r^2$ representing the circular edge is solved to obtain the boundary of ROI, (**B**), a circular ROI is then extracted, (**C**).
(TIF)

**S6 Fig. Detailed criteria for detecting ROIs containing cells.** The first test (**A**) reduces the number of clumped cells based on size and aspect ratio. From the passed images, images with shadow artefacts are removed based on the ratio of mean channel intensities of red and green channels, (**B**), which are then again filtered based on Hematoxylin stained area, (**C**). Finally, the ROIs are analysed to find the presence of a nucleus, (**D**).
(TIF)

**S7 Fig. Inception v3 architecture.** The deep convolutional neural network used here to delineate between normal cells and atypical cells.
(TIF)

**S1 Table. Showing demographics of subjects included in study.**
(DOCX)

**S2 Table. Machine learning model comparison.** Comparison of performance of various machine learning models used to delineate Oral Squamous Cell Carcinoma (OSCC) from High Grade Dysplasia (HGD) and Low-Grade Dysplasia (LGD).
(DOCX)

## Acknowledgments

We would like to acknowledge Thiyagarajan Subramani, Siemens Healthcare Pvt Ltd, Bangalore, Joachim Bangert, Siemens Healthcare GmbH, Germany and Radu Miron Toev, Siemens S.R.L, Romania for technical help.

## Author Contributions

**Conceptualization:** Arun Baby, Praveen Gurpur, Arunan Skandarajah, Felix Koch, Daniel A. Fletcher, Manohar Kollegal, Lance Ladic, Amritha Suresh, Hardik J. Pandya, Moni Abraham Kuriakose.

**Data curation:** Sumsum Sunny, Bonney Lee James, Ravindra Doddathimmasandra Ramanjinappa, Sunil Paramel Mohan, Nisheena Raghavan, Uma Kandasarma, Sangeetha N., Subhasini Raghavan, Naveen Hedne, Felix Koch, Daniel A. Fletcher, Praveen Birur N., Lance Ladic, Amritha Suresh.

**Formal analysis:** Sumsum Sunny, Arun Baby, Bonney Lee James, Dev Balaji, Aparna N. V., Praveen Gurpur, Michael D'Ambrosio, Nisheena Raghavan, Uma Kandasarma, Subhasini Raghavan, Naveen Hedne, Felix Koch, Sumithra Selvam, Praveen Birur N., Lance Ladic, Amritha Suresh, Hardik J. Pandya, Moni Abraham Kuriakose.

**Funding acquisition:** Praveen Gurpur, Lance Ladic, Amritha Suresh, Moni Abraham Kuriakose.

**Investigation:** Sumsum Sunny, Praveen Gurpur, Arunan Skandarajah, Ravindra Doddathimmasandra Ramanjinappa, Sunil Paramel Mohan, Sumithra Selvam, Manohar Kollegal, Praveen Birur N., Amritha Suresh, Hardik J. Pandya, Moni Abraham Kuriakose.

**Methodology:** Sumsum Sunny, Arun Baby, Bonney Lee James, Aparna N. V., Maitreya H. Rana, Arunan Skandarajah, Michael D'Ambrosio, Ravindra Doddathimmasandra Ramanjinappa, Sangeetha N., Sumithra Selvam, Praveen Birur N., Amritha Suresh, Moni Abraham Kuriakose.

**Project administration:** Sumsum Sunny, Praveen Gurpur, Michael D'Ambrosio, Manohar Kollegal, Praveen Birur N., Lance Ladic, Amritha Suresh, Hardik J. Pandya, Moni Abraham Kuriakose.

**Resources:** Praveen Gurpur, Arunan Skandarajah.

**Software:** Arun Baby, Dev Balaji, Aparna N. V., Praveen Gurpur, Arunan Skandarajah, Felix Koch, Daniel A. Fletcher, Manohar Kollegal, Lance Ladic.

**Supervision:** Praveen Gurpur, Subhasini Raghavan, Daniel A. Fletcher, Manohar Kollegal, Hardik J. Pandya, Moni Abraham Kuriakose.

**Validation:** Arun Baby, Dev Balaji, Sunil Paramel Mohan, Nisheena Raghavan, Uma Kandasarma, Hardik J. Pandya.

**Visualization:** Arun Baby, Dev Balaji, Maitreya H. Rana, Manohar Kollegal, Praveen Birur N.

**Writing – original draft:** Sumsum Sunny, Arun Baby, Bonney Lee James, Maitreya H. Rana, Arunan Skandarajah, Subhasini Raghavan, Felix Koch, Sumithra Selvam, Praveen Birur N.

**Writing – review & editing:** Manohar Kollegal, Lance Ladic, Amritha Suresh, Hardik J. Pandya, Moni Abraham Kuriakose.

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
