## [Decision Letter · Decision Letter 0]

15 Aug 2019

PONE-D-19-15321

A smart tele-cytology point-of-care platform for oral cancer screening

PLOS ONE

Dear Dr. Kuriakose,

Thank you for submitting your manuscript to PLOS ONE. After careful consideration, we feel that it has merit but does not fully meet PLOS ONE’s publication criteria as it currently stands. Therefore, we invite you to submit a revised version of the manuscript that addresses the points raised during the review process.

We would appreciate receiving your revised manuscript by Sep 29 2019 11:59PM. To enhance the reproducibility of your results, we recommend that if applicable you deposit your laboratory protocols in protocols.io, where a protocol can be assigned its own identifier (DOI) such that it can be cited independently in the future. For instructions see: http://journals.plos.org/plosone/s/submission-guidelines#loc-laboratory-protocols

We look forward to receiving your revised manuscript.

Kind regards,

Fernando Schmitt, MD, PhD

Academic Editor

PLOS ONE

Journal Requirements:

2. Thank you for including your ethics statement on the online submission form; "Narayana Health Medical Ethics Committee [NHH/MEC-CL-2014/222]. Written informed consent were obtained"

Please amend your current ethics statement to confirm that your named institutional review board or ethics committee specifically approved this study.

3. In the Methods, please report the sample size that was calculated for this study.

4. In your Methods section, please provide additional information about the participant recruitment method and the demographic details of your participants. Please ensure you have provided sufficient details to replicate the analyses such as: a) the recruitment date range (month and year), b) a description of any inclusion/exclusion criteria that were applied to participant recruitment, c) a table of relevant demographic details, d) a statement as to whether your sample can be considered representative of a larger population, e) a description of how participants were recruited, and f) descriptions of where participants were recruited and where the research took place.

5.  Thank you for stating the following in the Financial Disclosure section:

This work was supported by the Wellcome Trust/DBT India Alliance Fellowship [IA/RTF/15/1/1017] awarded to Sumsum Sunny

https://www.indiaalliance.org/fellowsprofile/dr-sumsum-sunny--270

We note that one or more of the authors are employed by a commercial company: Siemens Healthcare Pvt Ltd and Siemens Healthineers

6. Please include your tables as part of your main manuscript and remove the individual files. Please note that supplementary tables (should remain/ be uploaded) as separate "supporting information" files

Additional Editor Comments:

Please address the questions of the reviewer.

Reviewers' comments:

Reviewer's Responses to Questions

**Comments to the Author**

1. Is the manuscript technically sound, and do the data support the conclusions?

Reviewer #1: Yes

Reviewer #2: Yes

2. Has the statistical analysis been performed appropriately and rigorously? 

Reviewer #1: Yes

Reviewer #2: Yes

3. Have the authors made all data underlying the findings in their manuscript fully available?

Reviewer #1: Yes

Reviewer #2: Yes

4. Is the manuscript presented in an intelligible fashion and written in standard English?

Reviewer #1: Yes

Reviewer #2: Yes

5. Review Comments to the Author

Reviewer #1: The authors evaluated the efficacy of a tele-cytology platform for oral cancer screening and developed a

risk stratification model using an Artificial Neural Network. The study is very interesting and designed in detailed.

Reviewer #2: This article is very well written and demonstrates the potential of a practical approach using artificial intelligence to provide adequate assistance to PoC. It is clearly described. It would be useful for the reader to have the resolution of CellScope specified in pixels/micrometer as well as the optical resolution used. The former was not described in your previous study entitled: 'Mobile microscopy as a screening tool for oral cancer in India: A pilot study.' Such data would also help to highlight that cytologic diagnosis can be performed under lower optical resolutions.

6. PLOS authors have the option to publish the peer review history of their article (what does this mean?). If published, this will include your full peer review and any attached files.

Reviewer #1: No

Reviewer #2: No

---

## [Author Response · Author response to Decision Letter 0]

17 Oct 2019

Response to Editor’s/Reviewers Comments

We thank the Editor and the Reviewers for their comments. Their comments have enabled us to improve the manuscript. Please see the response to each comment listed below. 

Response to Editor’s comments

Please ensure that your manuscript meets PLOS ONE's style requirements, including those for file naming. The PLOS ONE style templates can be found at http://www.journals.plos.org/plosone/s/file?id=wjVg/PLOSOne_formatting_sample_main_body.pdf and http://www.journals.plos.org/plosone/s/file?id=ba62/PLOSOne_formatting_sample_title_authors_affiliations.pdf

Response: Thank you for reviewing the article. We have revised manuscript and corrected according to PLOS ONE’s style.

2. Thank you for including your ethics statement on the online submission form; "Narayana Health Medical Ethics Committee [NHH/MEC-CL-2014/222]. Written informed consent were obtained"

Please amend your current ethics statement to confirm that your named institutional review board or ethics committee specifically approved this study.

Response: Our current ethics committee has reviewed and approved the study. 

We have amended the ethics statement to include this in the revised manuscript. (Page 5; line 116). We have also made the change in the submission form.

3. In the Methods, please report the sample size that was calculated for this study.

Response: The minimum sample size for screening was calculated as previously established [1]. The prevalence of expected neoplastic and high grade dysplastic lesions were approximately 80% [2]. We expected sensitivity of 80% and with 15% improvement [2, 3], considering the alpha value of <0.05 and power of 80%, the minimum sample size required for the study was 63 including Low grade–OPML (n=7), High Grade- OPML (n=28) and OSCC (n=28). We expected a drop out of 50% due to lack of biopsy, poor images and less number of cells. Therefore 100 subjects were recruited for the study including OPML and Malignant lesions. 

This information has been included in the manuscript (Page 8; line 169). 

4. In your Methods section, please provide additional information about the participant recruitment method and the demographic details of your participants. 

Response: Please see the response to each comment below

 a) The recruitment date range (month and year)- 

Response: The patient recruitment period was October 2014 to September 2016. The details have been added in the revised manuscript (page 5; 116).

b) a description of any inclusion/exclusion criteria that were applied to participant recruitment

Response: Subjects who are clinically diagnosed with oral potentially malignant and malignant lesion and given consent for the study were included and subjects who are already undergone biopsy for the oral lesions were excluded from the study (page 5; 117).

c) a table of relevant demographic details, 

Response: Demographic details were added in revised manuscript as a supplementary table S1 (Table is referred to in the manuscript, Page 9; line 196).

d) a statement as to whether your sample can be considered representative of a larger population

Response: Subjects recruited for the study were largely from southern states of India, including Karnataka, Tamil Nadu, and Kerala. The sample can hence be considered representative of the South Indian population. (page 9; 187)

e) a description of how participants were recruited, and 

Response: The subjected who meet inclusion and exclusion criteria were consented for the study after the study protocol is explained in detail. The protocol, merits and confidentiality of the study was explained to all subjects in their native language. The consent form was provided to the patient in their respective language and was signed by them, their physician and the study investigator (Page 5; line 117, Page 9; line 189).

f) description of where participants were recruited and where the research took place.

Response: The study was carried out in subjects recruited from the out-patient clinics of Department of Oral Medicine and Radiology, KLES Institute of Dental Sciences, Bangalore and Head and Neck Oncology Department, Mazumdar Shaw Medical Center, Bangalore, India (Page 5; line 113). The pathology review of the slides/images was carried out at the department of pathology of the two centres. Image analysis, AI was carried out at Integrated head and neck oncology program, MSMF and at the Department Biomedical and Electronic Engineering Systems Laboratory, IISc. 

5. Thank you for stating the following in the Financial Disclosure section:

This work was supported by the Wellcome Trust/DBT India Alliance Fellowship [IA/RTF/15/1/1017] awarded to Sumsum Sunny https://www.indiaalliance.org/fellowsprofile/dr-sumsum-sunny--270

We note that one or more of the authors are employed by a commercial company: Siemens Healthcare Pvt Ltd and Siemens Healthineers

Response: The authors from Siemens Healthcare Pvt Ltd and Siemens Healthineers have supported the study by providing the instrument, Cellscope and IT support for the instrument and reviewed the final manuscript. They were also involved in managing the collaboration between the multiple centres. 

The funding statement has also been modified as “The funder provided support in the form of salaries for authors [DBT Alliance fellowship: SPS; Siemens Healthcare Pvt Ltd and Siemens Healthineers: LL, PG, MK], but did not have any additional role in the study design, data collection and analysis, decision to publish, or preparation of the manuscript. The specific roles of these authors are articulated in the ‘author contributions’ section.”

We have amended the Authors Contribution section on Online submission accordingly.

Response: The Competing interests statement now includes the statement “This does not alter our adherence to PLOS ONE policies on sharing data and materials”

Both the Competing Interests Statement and the Funding Statement have been included in the cover letter. 

6. Please include your tables as part of your main manuscript and remove the individual files. Please note that supplementary tables (should remain/ be uploaded) as separate "supporting information" files

Response: Thank you for the comment. We have added the tables in main manuscript and removed individual files. Supplementary files have been retained separately. 

Response to Reviewer's Comments

Reviewer #1: The authors evaluated the efficacy of a tele-cytology platform for oral cancer screening and a risk stratification model using an Artificial Neural Network. The study is very interesting and designed in detailed.

Response: We thank the Reviewer for reviewing the manuscript and for their positive comments. 

Reviewer #2: This article is very well written and demonstrates the potential of a practical approach using artificial intelligence to provide adequate assistance to PoC. It is clearly described. It would be useful for the reader to have the resolution of CellScope specified in pixels/micrometer as well as the optical resolution used. The former was not described in your previous study entitled: 'Mobile microscopy as a screening tool for oral cancer in India: A pilot study.' Such data would also help to highlight that cytologic diagnosis can be performed under lower optical resolutions.

 Response: Thank you for reviewing the article and for all encouraging comments. We agree that information regarding the resolution will be important for the reader. The resolution of Cellscope is 1.9 pixel/micrometre (optical resolution is 200x). 

We have added this in revised manuscript (Page 6; line 121). 

References

1. Bujang MA, Adnan TH. Requirements for Minimum Sample Size for Sensitivity and Specificity Analysis. Journal of clinical and diagnostic research : JCDR. 2016;10(10):YE01-YE6. doi: 10.7860/JCDR/2016/18129.8744. PubMed PMID: 27891446; PubMed Central PMCID: PMC5121784.

2. Skandarajah A, Sunny SP, Gurpur P, Reber CD, D'Ambrosio MV, Raghavan N, et al. Mobile microscopy as a screening tool for oral cancer in India: A pilot study. PloS one. 2017;12(11):e0188440. doi: 10.1371/journal.pone.0188440. PubMed PMID: 29176904; PubMed Central PMCID: PMC5703562.

3. Hajian-Tilaki K. Sample size estimation in diagnostic test studies of biomedical informatics. Journal of biomedical informatics. 2014;48:193-204. doi: 10.1016/j.jbi.2014.02.013. PubMed PMID: 24582925.

---

## [Editor Report · Decision Letter 1]

24 Oct 2019

A smart tele-cytology point-of-care platform for oral cancer screening

PONE-D-19-15321R1

Dear Dr. Kuriakose,

We are pleased to inform you that your manuscript has been judged scientifically suitable for publication and will be formally accepted for publication once it complies with all outstanding technical requirements.

With kind regards,

Fernando Schmitt, MD, PhD

Academic Editor

PLOS ONE
---

## [Editor Report · Acceptance letter]

31 Oct 2019

PONE-D-19-15321R1 

A smart tele-cytology point-of-care platform for oral cancer screening 

Dear Dr. Kuriakose:

I am pleased to inform you that your manuscript has been deemed suitable for publication in PLOS ONE. Congratulations! Your manuscript is now with our production department. 

With kind regards,

on behalf of

Prof Fernando Schmitt 

Academic Editor

PLOS ONE